# Determinants of hypertension among patients with type 2 diabetes mellitus on follow-up at Tikur Anbessa Specialized Hospital, Addis Ababa: A case-control study

**Kehabtimer Shiferaw Kotiso**[1,2]*, **Nabiha Degemu**[2], **Samson Gebremedhin**[3],
**Melaku Taye**[4], **Adane Petros**[4], **Fanuel Belayneh**[2], **Deneke Wolde**[5], **Dejene Hailu**[2]

1 Department of Public Health, College of Medicine and Health Sciences, Werabe University, Werabe, Ethiopia, 2 School of Public Health, College of Medicine and Health Sciences, Hawassa University, Hawassa, Ethiopia, 3 School of Public Health, College of Health Sciences, Addis Ababa University, Addis Ababa, Ethiopia, 4 School of Medicine, College of Health Sciences, Addis Ababa University, Addis Ababa, Ethiopia, 5 Department of Medical Laboratory, College of Medicine and Health Sciences, Wachemo University, Hossana, Ethiopia

* kehabtimershfrw@gmail.com

**Data Availability Statement:** All relevant data are within the manuscript and its Supporting Information files.

# Abstract

## Introduction

Hypertension (HTN) in patients with diabetes mellitus (DM) is a common problem that increases the risk of mortality and morbidity, and lowers the quality of life. Despite the disproportionately high burden of HTN in DM patients, determinants for the comorbidity have not been sufficiently explored. Therefore, this study aimed to identify the determinants of HTN among patients with type 2 diabetes mellitus on follow-up at Tikur Anbessa Specialized Hospital.

## Methods and materials

We conducted a hospital-based unmatched case-control study at Tikur Anbessa Specialized Hospital on 386 randomly selected patients with type 2 diabetes on follow-up (200 cases and 186 controls). We collected data by using a structured interviewer-administered questionnaire and data extraction form. To identify determinants of hypertension, a multivariable binary logistic regression was fitted, and the findings are presented using adjusted odds ratio (AOR) with 95% confidence interval (CI).

## Results

The mean reported age (±SD) of the cases and the controls was 60.3 (±9.9) and 55.3 (±11.3) years, respectively. The eight identified independent determinants of hypertension with AOR [95% CI] were obesity: 2.82 [1.43, 5.57], sedentary activity of ≥4hrs/day: 1.75 [1.10, 2.79], higher stress score: 1.05 [1.01, 1.10], serum creatinine above 1.1 mg/dl: 2.35 [1.13, 4.91], age: 1.05 [1.02, 1.08], being government employee as compared to private workers: 2.18 [1.06, 4.50] and family history of hypertension: 2.11 [1.26, 3.54]. Further,

**Funding:** Hawassa University (https://www.hu.edu.et/) funded this study, and Kehabtimer Shiferaw received the award. The funders had no role in study design, data collection and analysis, decision to publish, or preparation of the manuscript.

**Competing interests:** The authors have declared that no competing interests exist.

interaction of diabetes duration with insulin use: 1.03 [1.01, 1.07] was also a significant predictor of HTN among DM patients.

## Conclusion

The finding calls for interventions for mitigating these determinants. Further research is needed to examine the interaction between diabetes duration and insulin use.

## Background

Diabetes mellitus (DM) is a chronic disease that occurs due to insufficient insulin production or ineffective insulin utilization by our body. Based on the International Diabetes Federation (IDF) 2019 estimate, globally 463 million adults were affected by DM, of whom four-fifths live in low- and middle-income countries (LMICs). The figure is further projected to increase to 578.4 million by 2030 and 700.2 million by 2045. Among its different types, type 2 diabetes is the most common, accounting for around 90% of all cases [1]. According to a recent analysis, the pooled prevalence of type 2 diabetes mellitus (T2DM) in Ethiopia is about 5% [2].

Globally, premature death attributable to diabetes and its complications is rising at an alarming rate with an estimated death exceeding four million in 2019 [1, 3, 4]. The percentage of premature deaths attributable to high blood glucose is higher in LMICs than in high-income countries (HICs) [4]. In Ethiopia, in 2019 there were about 1.7 million cases and 23,157 deaths in adults due to diabetes [1]. The DM incidence is also growing at an alarming rate [5].

Hypertension (HTN) is the most common comorbidity among people with type 2 diabetes [2]. Its prevalence in this population is sharply increasing in all regions of the world with a prevalence of 50 to 75% in most studies [6]. Even though limited studies are available in Ethiopia, a study reported a 55% prevalence of hypertension among DM patients on follow-up in a referral hospital [7].

Diabetes, HTN, or a combination of both, causes 80% of end-stage renal disease globally [1]. The combination of HTN and type 2 diabetes significantly increases the risk of cardiovascular events [8]. Besides, the comorbidity increases the risk of resistant hypertension [9], kidney disease, including nephropathy [1], diabetic peripheral neuropathy [10], retinopathy, depression, lower quality of life, and health care costs [11–13]. HTN coexistence in diabetics is a major contributor to the development and progression of micro- and macro-vascular complications [14, 15], which are the leading causes of morbidity and mortality among DM cases [16–18]. Moreover, the co-morbidity considerably impairs health-related quality of life (HRQoL) [19, 20].

For people with T2DM, comprehensive cardiovascular risk assessment, counseling and management are an essential part of diabetes management [21]. This is because the majority (>68%) of diabetic patients dies due to cardiac complications and 16% die due to stroke. In addition to the already existing excess risk of Cardiovascular disease (CVD) death in diabetic patients, HTN incurs an additional risk of CVD-related death in patients with T2DM [22]. Since HTN is one of the prominent risk factors for CVD in this population, exploring its determinants is substantive for the prevention and control of CVD and other DM complications [18].

Preventing HTN is more advantageous than treating it, particularly among DM cases [1]. HTN would lead to systemic malfunctioning which is difficult to manage [23]. The prevention also conforms with the World Health Organization (WHO) Global Action Plan, which aimed

to achieve a 25% relative reduction in HTN prevalence as one of its three aims [24]. However, further evidence is required on the predictors of HTN for the successful prevention of the disease and its deleterious consequences especially among the disproportionately affected group i.e., the diabetic population, in whom the risk factors for HTN might be different from the general population.

Lowering HTN is among the recommended measures by the IDF to significantly reduce the risk of CVD outcomes and chronic kidney disease [1]. The need for a reduction of HTN burden among DM patients is extensively studied; however, the way how we reduce it in this specific population demands further evidence. Despite the disproportionately high burden of HTN, as per our knowledge, there are only three studies [7, 25, 26] conducted on this specific population. These few studies have major methodological limitations, including flawed participant population selection to explore determinant factors for the development of HTN among T2DM patients. Therefore, this study addressed the aforementioned issues and identified determinants of HTN among T2DM patients that can be used for the effective prevention and control of the condition.

## Methods and materials

### Study setting, design and period

This institution-based unmatched case-control study was conducted at the Diabetic clinic of Tikur Anbessa Specialized Hospital (TASH) in Addis Ababa, the capital of Ethiopia, from March 01 to June 30, 2020. TASH is a tertiary referral hospital for entire Ethiopia. It provides specialty, sub-specialty, and super-specialty health care services.

The diabetic clinic is one of the centers of the hospital, which is designated for care and follow-up of patients with diabetes. The clinic provides services for the patients on all working days of the week with a daily average of 70 to 80 clients.

### Population

Cohorts of T2DM patients on follow-up in TASH diabetic clinic were used as the source population for this study. The study population were T2DM patients with follow-up appointments at the TASH diabetic clinic scheduled between March and June 2020. T2DM patients who had at least one prior visit at TASH diabetic clinic and age of 18 years and above were included in the study. Pregnant women and patients whose hypertension diagnosis precede DM diagnosis were excluded from the study.

T2DM patients with comorbid hypertension were considered as cases. On the other hand, T2DM patients with no hypertension were taken as controls.

### Sample size determination and sampling procedure

The sample size was calculated using Stata version 14.1 with the assumptions of 95% confidence level, 90% power, OR &= 2.28 [7], case to control ratio of 1:1, and proportion of controls with an exposure of 70%. Considering a non-response rate of 10%, the final sample size of 202 cases and 202 controls was reached.

We used a simple random sampling technique to select participants. A digital record of the hospital, where the status and care of DM patients are regularly recorded per each visit, was used as a sampling frame and computer-generated random numbers were used to select the study participants. To illustrate, whether the patient has comorbidity including hypertension or not is recorded in the digital record, and all DM patients be screened for hypertension per each visit. As a result, those T2DM patients already recorded as hypertensive in the record

were considered as list of cases, and the non-hypertensives as controls to apply the random sampling. In the process, the newly diagnosed cases were also managed accordingly.

## Data collection techniques

The questionnaire was prepared in English and translated into Amharic. Finally, it was translated back to English to check its consistency. Pretest was conducted on 41 T2DM patients who were not included in the study population at TASH before actual data collection. Open Data Kit (ODK) version 1.25.2 software [https://opendatakit.org/] was used for data collection along with the KoboToolbox server to store the collected data. Four BSc nurses were recruited for the data collection. Training on how to use the software was given to the data collectors.

Data on socio-demographic and behavioral factors were collected through a face-to-face interview using "WHO STEPS Instrument for Chronic Disease Risk Factor Surveillance" [27]. To assess the stress scale, validated Cohen's [28] 10 item perceived stress scale (PSS) [29] was adapted and used. To obtain PSS scores, the response to the four positively stated items were reverse coded and then all the scale items were summed up.

Data on clinical and laboratory profiles of the participants were obtained through a review of digital records of the participants using a data extraction tool. The biological parameters obtained through a review of digital record were fasting blood sugar (FBS), total cholesterol, low-density lipoprotein (LDL), high-density lipoprotein (HDL), triglyceride, serum creatinine, hemoglobin A1c (HA1C), systolic blood pressure (SBP), and diastolic blood pressure (DBP). Besides, anthropometric data, including BP were collected by direct measurement as described below.

## Measurements

Blood pressure was measured using a mercury sphygmomanometer (adult size) following standard procedure. The sphygmomanometer was used based on the 2018 consensus document of a Universal Standard for the Validation of Blood Pressure Measuring Devices [30]. BP was taken in a sitting position from the left arm with feet on the floor and arm supported at heart level after the patient rested for at least 5 minutes. Those patients who have taken caffeine during the hour preceding the measurement and those who have smoked during the preceding 30 minutes were let to stay for at least 1 hour and 30 minutes respectively from the time of those events. To reduce within-patient variability, 2 measurements of the last consecutive visits at least 14 days apart were taken, and the average value was used for the analysis [31, 32]. Hypertension was defined as an elevated average BP of 2 measurements made at subsequent visits of at least 14 days apart (SBP ≥140 mmHg and/or DBP≥90 mmHg) or taking antihypertensive medication.

Physical activity was measured using the Global Physical Activity questionnaire included in the WHO Steps questionnaire. Exercise adherence was defined as a brisk walk for 30 to 40 minutes or more in 4 or more days of a week, and/or a minimum of 150 minutes per week of moderate-intensity and/or 90 minutes per week of vigorous cardio-respiratory exercise. Individuals were considered to have sedentary activity if they report spending ≥ 4hrs by sitting or reclining at work, home, during transportation, and or with friends but do not include the time spent by sleeping.

The weight of the participants was taken using calibrated beam balance and the scale was checked to zero before each measurement. Participants' weight was measured after removing heavy clothes and recorded to the nearest 0.1 kg. Height was measured using the standard measuring scale and method. The occiput, shoulder, buttocks, and heels touched the measuring board, and height was recorded to the nearest 0.1cm. To compute body mass index (BMI),

height and weight were automatically calculated using ODK as BMI = weight in kg/ (height in m)$^2$.

## Data quality and management

To ensure the quality of data and reduce intra and inter observation difference on the measurement of variables, a pre-test was conducted on 41 people with type 2 diabetes at TASH, and training was given to data collectors in Addis Ababa for one day before the survey. Excel template for the ODK was prepared with relevant restrictions and necessary commands and tested before actual data collection. The collected data were checked for completeness and consistency daily. Regular supervision and monitoring were made by the assigned supervisors and the principal investigator.

## Data processing and analysis

The ODK collected data were validated and exported to Stata version 14.0 for analysis. The complete data set used in the analysis is provided with "S1 Dataset". Mean, median, standard deviation, interquartile range, and proportion were used to describe the data.

To identify determinants of hypertension among people with T2DM, a binary logistic regression analysis was done. All important variables were initially analyzed using a bivariable analysis. Theoretically important confounders, irrespective of their P-value and variables with a P-value < 0.25 in a bivariable analysis were included in the multivariable model to control for confounders. Akaike's information criteria (AIC) and Bayesian information criteria (BIC) were used to select the best-fitting model, and finally, a model with the least AIC and BIC was used as a final model. All the preliminary assumptions of the model such as multi-collinearity and model fitness were checked. Those variables showing multi-collinearity were removed from the model and the p-value of the Hosmer Lemeshow was found to be 0.25. Lastly, variables with a p-value ≤ 0.05 in the multivariable analysis were considered statistically significant, and AOR with 95% CI was estimated to measure the strength of the associations. The results are described using text, tables, and graphs, and finally interpreted into valuable information.

## Ethical consideration

Ethical clearance was obtained from the Institutional Review Board of Hawassa University College of Medicine and Health Sciences. Informed verbal consent was obtained from the participants without any inducement, undue influence, or coercion. Confidentiality was maintained at all levels of the study. Besides, participants who were found to have a high BP were given education on how to control BP and linked to their care providers.

## Results

### Characteristics of the respondents

In this study, a total of 200 cases with both type 2 diabetes and hypertension and 186 controls with only type 2 diabetes, but no hypertension were participated. The mean (±SD) age was 60.3 (± 9.9) years for cases and 55.3 (±11.3) years for controls. Among the participants, almost half of the cases and controls (47.3%) were females, and nearly all of the respondents (96.0%) were urban residents. The median (IQR) reported duration of DM from the diagnosis was 15 (9, 20) years in cases and 10 (6, 17) years in controls. One-third of the cases and about a quarter of the controls reported a family history of HTN. Over two-thirds (68.5%) of the cases and a half (50.5%) of the controls reported using insulin. This study also revealed that there is a

sedentary activity of ≥4hour/day in the majority (62.0%) of the cases and in less than half (43.0%) of the controls (Table 1). Of the study participants, 23.5% of the cases and 12.9% of the controls were obese (Fig 1).

## Metabolic risk factors of hypertension

The median SBP (Q1, Q3) and DBP (Q1, Q3) of the cases were 135 (126, 145) mmHg and 80 (75, 85) mmHg, respectively. However, the median SBP (Q1, Q3) and DBP (Q1, Q3) of the controls were 125 mmHg (120, 132) and 80 (70, 80) mmHg, respectively. (Table 2)

## Determinants of hypertension among people with T2DM

Based on the p-value of the bivariable analyses, sixteen variables were identified as candidate variables for the multivariable model. These were age, occupation, duration of DM since diagnosis, diabetes medication, the interaction of duration of DM with insulin, having glucometer, family history of hypertension, visiting traditional healer, adherence to exercise, sedentary activity, stress score, BMI, retinopathy, nephropathy, serum creatinine level, and smoking status. Diabetes medications and duration of DM were excluded from the final model due to multi-collinearity.

The result of the multivariable analysis identified obesity, sedentary activity, stress scores, the interaction of diabetes duration with insulin use, serum creatinine of >1.1 mg/dl, age, government employee, and family history of hypertension as independent determinants of hypertension among people with T2DM. (Table 3).

## Discussion

Obesity was identified as one of the significant determinants of hypertension in people with T2DM. The odds of being obese (BMI > 30 kg/$m^2$) rather than normal (BMI <25/$m^2$) was about 3 times higher among cases than the controls. This is in line with cross-sectional studies conducted in Botswana [33], United Arab Emirates [34], Benghazi [35], Morocco [36], and Hossana, Ethiopia [26]. The association of obesity with hypertension might be due to adipocytes in obese individuals leading to the activation of the angiotensinogen which again increases sodium reabsorption and volume overload in the renal system [37]. The other mechanism by which adiposity in obese individuals leads to hypertension is due to the increased free fatty acids, which can cause HTN either through its effect on the renal system or through causing inflammation. The inflammation again causes endothelial dysfunction leading to arterial stiffness and vasoconstriction which can cause hypertension [37, 38].

The sedentary activity state was also identified as an independent determinant of hypertension among people with T2DM. The odds that the cases stay for 4 or more hours per day in a sedentary state was about 2 times higher than that of the controls. The finding is supported by the SUN cohort [39] and a cross-sectional study conducted at Hossana NEMM hospital [26]. It also supports the country's NSAP for the prevention of NCD, which prioritized health promotion and disease prevention targeting behavioral risk factors. Therefore, it implies that this effort still needs to be strengthened and continued [40]. The possible justification for the association of sedentary activity with hypertension might be the fact that people who stay in a sedentary state have less physical activity level, which exposes them to unhealthy weight gain which leads to hypertension [39].

This study detected an interaction of diabetes duration with insulin use to be associated with hypertension among people with T2DM. The percent odds for each 5-year increase in the duration of DM when they use insulin was about 18% higher for cases than the controls. It suggests that insulin use alone might not be associated with hypertension, but depends on the

**Table 1.  Characteristics of people with T2DM on follow-up at TASH hospital, Addis Ababa, Ethiopia, 2020.**

| Variables | Cases, no. (%) | Controls, no. (%) | Total, no. (%) |
|---|---|---|---|
| **Sex (n = 386)** | | | |
| Male | 100 (50.0%) | 88 (47.3%) | 188 (48.7%) |
| Female | 100 (50.0%) | 98 (52.7%) | 198 (51.3%) |
| **Residence (n = 386)** | | | |
| Urban | 193 (96.5%) | 180 (96.8%) | 373 (96.6%) |
| Rural | 7 (3.5%) | 6 (3.2%) | 13 (3.4%) |
| **Current marital status (n = 386)** | | | |
| Single | 7 (3.5%) | 8 (4.3%) | 15 (3.9%) |
| Married | 136 (68.0%) | 141 (75.8%) | 277 (71.8%) |
| Divorced | 24 (12.0%) | 16 (8.6%) | 40 (10.4%) |
| Widowed | 33 (16.5%) | 21 (11.3%) | 54 (14.0%) |
| **Educational status (n = 386)** | | | |
| No formal education | 14 (7.0%) | 14 (7.5%) | 28 (7.3%) |
| Primary education | 42 (21.0%) | 45 (24.2%) | 87 (22.5%) |
| Secondary education | 64 (32.0%) | 55 (29.6%) | 119 (30.8%) |
| Above secondary education | 80 (40.0%) | 72 (38.7%) | 152 (39.4%) |
| **Occupation (n = 386)** | | | |
| Private work | 33 (16.5%) | 49 (26.3%) | 82 (21.2%) |
| Government employee | 36 (18.0%) | 35 (18.8%) | 71 (18.4%) |
| Housewife | 68 (34.0%) | 60 (32.3%) | 128 (33.2%) |
| Unemployed | 7 (3.5%) | 6 (3.2%) | 13 (3.4%) |
| Retired | 56 (28.0%) | 36 (19.4%) | 92 (23.8%) |
| **Household monthly income (n = 386)** | | | |
| </ = 1000 ETB | 54 (27.0%) | 42 (22.6%) | 96 (24.9%) |
| 1001–5000 ETB | 114 (57.0%) | 116 (62.4%) | 230 (59.6%) |
| >5000 ETB | 32 (16.0%) | 28 (15.1%) | 60 (15.5%) |
| **Family history of hypertension (n = 386)** | | | |
| Yes | 66(33.0%) | 44(23.7%) | 110(28.5%) |
| No | 127(63.5%) | 141(75.8%) | 268(69.4%) |
| **Ever used diabetes medication (n = 386)** | | | |
| Metformin | 153(76.5%) | 152 (81.7%) | 305(79.0%) |
| Glibenclamide | 93(46.5%) | 76(40.9%) | 169(43.8%) |
| Insulin | 137(68.5%) | 94(50.5%) | 231(59.8%) |
| **Missed dose in the last month (n = 386)** | | | |
| No | 181(90.5%) | 161(85.6%) | 342(88.6%) |
| 1–2 | 12(6%) | 10(5.4%) | 22(5.7%) |
| ≥3 | 7(3.5%) | 15(8.1%) | 22(5.7%) |
| **Ever visited traditional healers (n = 386)** | | | |
| Yes | 6(3.0%) | 10(5.4%) | 16(4.1%) |
| No | 194(97.0%) | 176(94.6%) | 370(95.9%) |
| **Current herbal medicine use (n = 16)** | | | |
| Yes | 2(33.3%) | 7(70.0%) | 9(56.3%) |
| No | 4(66.7%) | 3(30.0%) | 7(43.8%) |
| **Attend health education on Diabetes** | | | |
| Yes | 143(71.5%) | 127(68.3%) | 270(69.9%) |
| No | 57(28.5%) | 59(31.7%) | 116(30.1%) |
| **Member of diabetes association** | | | |

*(Continued)*

**Table 1.** (Continued)

| Variables | Cases, no. (%) | Controls, no. (%) | Total, no. (%) |
|---|---|---|---|
| Yes | 50(25.0%) | 38(20.4%) | 88(22.8%) |
| No | 150(75.0%) | 148(79.6%) | 298(77.2%) |
| **Have glucometer at home** | | | |
| Yes | 144(72.0%) | 120(64.5%) | 264(68.4%) |
| No | 56(28.0%) | 66(35.5%) | 122(31.6%) |
| **Days, in which glucose was measured/wk** | | | |
| Not measured at all | 45(22.5%) | 36(19.4%) | 81(21.0%) |
| 1–2 days | 99(49.5%) | 111(59.7%) | 210(54.4%) |
| 3 or more days | 56(28.0%) | 39(21.0%) | 95(24.6%) |
| **Hemoglobin A1c (n = 195)** | | | |
| <7% | 19 (21.3%) | 23 (21.7%) | 42(21.5%) |
| >7% | 70 (78.7%) | 83 (78.3%) | 153 (78.5%) |
| **Creatinine level (n = 386)** | | | |
| >1.1 mg/dl | 97(48.5%) | 103(55.4%) | 200 (51.8%) |
| ≤1.1mg/dl | 39 (19.5%) | 18(9.7%) | 57 (14.8%) |
| not determined | 64 (32.0%) | 65 (34.9%) | 129 (33.4%) |
| **Smoking** | | | |
| Ever smoked tobacco products (n = 386) | | | |
| Yes | 17(8.5%) | 16 (8.6%) | 33(8.5%) |
| No | 183 (91.5%) | 170(91.4%) | 353(91.5%) |
| **Alcohol consumption** | | | |
| Ever drunk alcoholic drinks (n = 386) | 78(39.0%) | 74(39.8%) | 152(39.4%) |
| Drank alcoholic drinks at least once a month in the last year (n = 152) | 12 (15.4%) | 12 (16.2%) | 24 (15.8%) |
| ≥ 3 alcoholic drinks per occasion (n = 152) | 17 (21.8%) | 14 (18.9%) | 31 (20.4%) |
| **Vegetable consumption per week (n = 386)** | | | |
| <4 servings | 72 (36%) | 77 (41.4%) | 149 (38.6) |
| >/ = 4 servings | 128 (64.0%) | 109 (58.6%) | 237 (61.4) |
| **Fruit consumption per week (n = 386)** | | | |
| <4 servings | 156 (78.0%) | 139 (74.7%) | 295 (76.4%) |
| >/ = 4 servings | 44 (22.0%) | 47 (25.3%) | 91 (23.6%) |
| **Type of oils (n = 386)** | | | |
| Vegetable oil | 194(97.0%) | 174(93.5%) | 368(95.3%) |
| Cholesterol oil and others | 6(3%) | 12(6.5%) | 18(4.7%) |
| **Salt consumption after diabetes (n = 386)** | | | |
| No change from predictable period | 16(8.0%) | 40(21.5%) | 56(14.5%) |
| Minimally decreased | 48(24.0%) | 41(22.0%) | 89(23.1%) |
| Substantially decreased | 101(50.5%) | 86(46.2%) | 187(48.4%) |
| Stopped at all | 35(17.5%) | 19(10.2%) | 54(14.0%) |
| **Sleeping duration per day (n = 386)** | | | |
| <7hrs | 45(22.5%) | 43(23.1%) | 88(22.8%) |
| ≥7hrs | 155(77.5%) | 143(76.9%) | 298(77.2%) |
| **Moderate intensity exercise (at recreation or work) (n = 386)** | | | |
| No | 185(92.5%) | 159(85.5%) | 344(89.1%) |
| Yes | 15(7.5%) | 27(14.5%) | 42(10.9%) |
| **Vigorous intensity exercise (at recreation or work) (n = 386)** | | | |
| No | 182(91.0%) | 160(86.0%) | 342(88.6%) |
| Yes | 18(9.0%) | 26(14.0%) | 44(11.4%) |

(*Continued*)

**Table 1.** (Continued)

| Variables | Cases, no. (%) | Controls, no. (%) | Total, no. (%) |
|---|---|---|---|
| **Doing Moderate and/or vigorous exercise (n = 386)** | | | |
| Neither | 171(85.6%) | 143(76.9%) | 314(81.3%) |
| Either | 25(12.5%) | 33(17.7%) | 58(15.0%) |
| Both | 4(2.0%) | 10(5.4%) | 14(3.6%) |
| **Walking or use of bicycle (minutes/week) (n = 386)** | | | |
| ≤120 | 81 (40.5%) | 86 (46.2%) | 167(43.3%) |
| >120 | 119 (59.5%) | 100(53.6%) | 219(56.7%) |
| **Sedentary activity (hrs/day) (n = 386)** | | | |
| <4 | 76(38.0%) | 106(57.0%) | 182(47.2%) |
| ≥4 | 124(62.0%) | 80(43.0%) | 204(52.8%) |

duration of diabetes as well. Though this interaction needs further investigation, it may be partly explained by the fact that type 2 diabetic patients who were on insulin might have experienced poor glycemic control probably due to insulin resistance [41]. The synergistic effect of insulin resistance, insulin-induced weight gain, and increased diabetic duration might have contributed to the development of hypertension in this population. It might also be explained by the already established risk among the insulin users just before the initiation of insulin other than the metformin users because both the patients and the physicians are not eager to start insulin on time [42]. They are, therefore, more likely to start insulin after the treatment failure [43]. The duration of treatment failure is even more pronounced in resource-limited

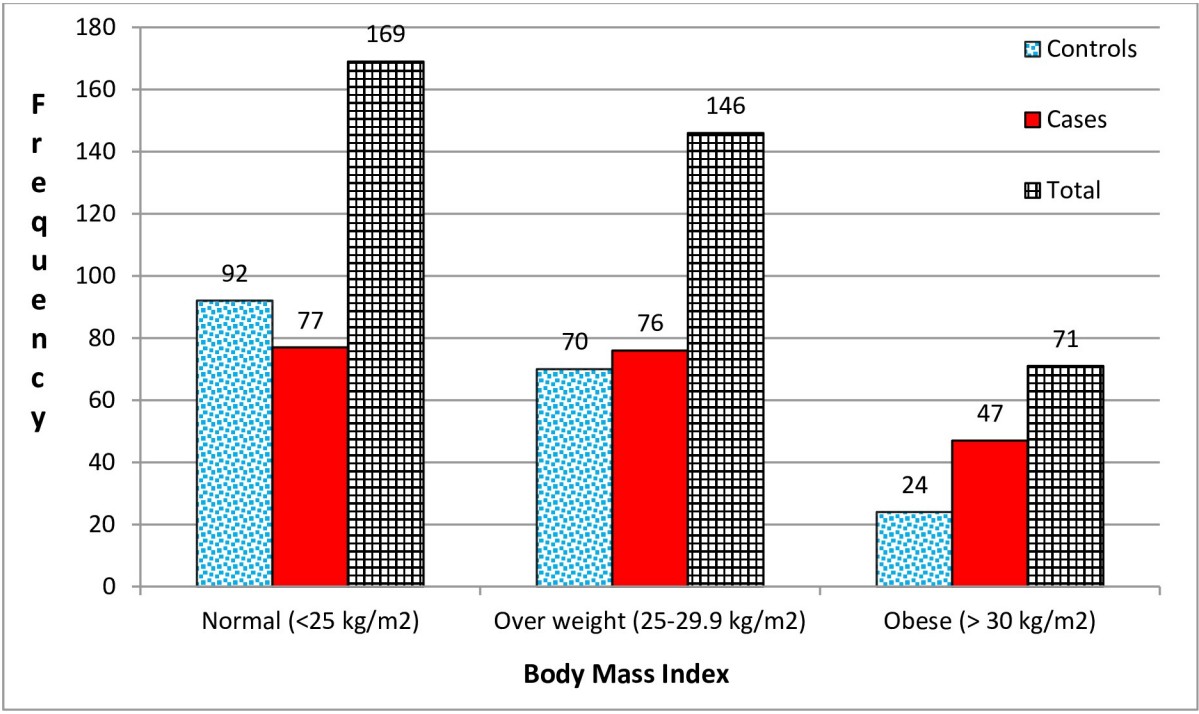

**Fig 1. Distribution of hypertensions by BMI status among people with T2DM on follow-up at TASH hospital, Addis Ababa, Ethiopia, 2020.**

**Table 2. Metabolic factors of people with T2DM on follow-up at TASH hospital, Addis Ababa, Ethiopia, 2020.**

| Variables | Cases | | Controls | |
|---|---|---|---|---|
| | Median | IQR (Q1, Q3) | Median | IQR (Q1, Q3) |
| Mean FBS in mg/dl (n = 386) | 145.5 | 126, 170 | 150 | 129, 176 |
| Total cholesterol in mg/dl (n = 212) | 164.5 | 128, 212 | 177 | 141, 216 |
| LDL cholesterol in mg/dl (n = 213) | 106.0 | 79.0, 135 | 111 | 82, 139 |
| HDL cholesterol in mg/dl (n = 213) | 41.0 | 33, 51 | 42 | 36, 49 |
| Triglyceride in mg/dl (n = 215) | 144.0 | 112, 200 | 150 | 106, 193 |
| Creatinine in mg/dl (n = 257) | 0.9 | 0.7, 1.2 | 0.8 | 0.6, 1.1 |
| Hemoglobin A1c % (n = 195) | 8.5 | 7.3, 10.1 | 8.6 | 7.2, 10.0 |
| Mean SBP in mmHg (n = 386) | 135 | 126, 145 | 125 | 120, 132 |
| Mean DBP in mmHg (386) | 80 | 70, 80 | 80 | 70, 80 |

**Key:** FBS = Fasting blood sugar, LDL = Low density lipoprotein, HDL = High density lipoprotein

countries such as Ethiopia due to poor health system that leads to long follow up intervals and delay with the treatment failure.

Stress score was also positively associated with hypertension in the study population. For each one-unit increase in the stress score, the risk of being hypertensive increases by 5%. The finding is supported by the study conducted to assess the role of stress in newly diagnosed T2DM and HTN by Kaur et al, who reported a positive linear trend between hypertension and stress [44]. The various hormone release during the stressful condition increases the BP subsequently leading to hypertension. The stimulation of the nervous system during stress produces large amounts of vasoconstricting hormones that increase BP. Moreover, the coupling of one risk factor with other stress-producing factors multiplies the effect on BP [45].

The other significant determinant of hypertension was occupational status. The odds of being hypertensive was 2 times higher among government employees than among private workers. This might reflect an indirect difference in the lifestyle and other risk factors of HTN between the two groups [39]. This finding is discordant with the studies conducted in Benghazi [35], Debre Tabor general hospital [46], and Hosanna NEMM hospital in Ethiopia [7, 26]. The reason for the disagreement might be the difference in the characteristics of the study population, small sample size, and failure to control for the confounding effect of stress in the previous studies which might have masked the association.

Likewise, age was identified as one of the independent risk factors contributing to the development of hypertension among people with T2DM. For each one-year increase in age, the percent odds that the patient develops hypertension increases by 5%. This is in agreement with the studies conducted in the United Arab Emirates [34], Benghazi [35], Botswana [33], Morocco [36], and Jordan [47]. Similarly, it agrees with the previous studies conducted at Hawassa University Comprehensive and Specialized Hospital [48] and two cross-sectional studies conducted in Hossana NEMM hospital, Southern Ethiopia [7, 26]. The association of age with hypertension could be due to age-induced changes in arterial and arteriolar stiffness. Large artery stiffness (LAS) is mainly due to arteriosclerotic structural alterations and calcification [49–51].

Serum creatinine level of >1.1 mg/dl was the other associated factor, which increased the risk of HTN by 2 folds among cases compared to controls. The elevation of creatinine level is an indicator of renal dysfunction leading to increased sodium reabsorption. Thus, the sodium loading may increase BP only when renal sodium excretion is constrained by ablation of 70% of renal mass or administration of angiotensin or aldosterone. Consequently, the expansion of

**Table 3. Bivariable and multivariable logistic regression analyses results of factors associated with hypertension among people with T2DM on follow-up at TASH hospital, Addis Ababa, Ethiopia, 2020.**

| Explanatory Variables | Cases | Controls | COR 95% CI | AOR 95% CI |
|---|---|---|---|---|
| Age | 60.3±9.9$^€$ | 55.0±11.3$^€$ | 1.05(1.03, 1.07) | 1.05(1.02,1.08)* |
| **Occupation** | | | | |
| Private work | 33 (16.5) | 49 (26.3) | 1 | 1 |
| Government employee | 36 (18.0) | 35 (18.8) | 1.53(0.80,2.90) | 2.18(1.06,4.50)* |
| Housewife | 68 (34.0) | 60 (32.3) | 1.68(0.96,2.95) | 1.27(0.65,2.46) |
| Unemployed | 7 (3.5) | 6 (3.2) | 1.73(0.53,5.61) | 1.76(0.44,7.00) |
| Retired | 56 (28.0) | 36 (19.4) | 2.31(1.26,4.24) | 1.22(0.57,2.60) |
| **Having Glucometer** | | | | |
| Yes | 144(72.0) | 120(64.5) | 1 | 1 |
| No | 56(28.0) | 66(35.5) | 0.71(0.46,1.09) | 0.85(0.52, 1.40) |
| **DM duration\*DM drug #** | | | | |
| Metformin alone | # | # | 1.01(0.95,1.06) | 1.00 (0.94, 1.06) |
| Glibenclamide | # | # | 1.03(0.99,1.07) | 1.01(0.97, 1.05) |
| Insulin | # | # | 1.05(1.03,1.09) | 1.03(1.01, 1.07)* |
| **Family history of HTN** | | | | |
| Yes | 66(33.0) | 44(23.7) | 1.59 (1.01,2.49) | 2.11(1.26,3.54)* |
| No | 127(63.5) | 141(75.8) | 1 | 1 |
| **Ever visited traditional healers** | | | | |
| Yes | 6(3.0) | 10(5.4) | 1 | 1 |
| No | 194(97.0) | 176(94.6) | 1.84(0.64,5.16) | 2.23(0.64,7.75) |
| **Exercise adherence** | | | | |
| Yes | 122(61.0) | 113(60.7) | 1.01(0.67,1.52) | 1.20(0.74,1.93) |
| No | 78(39.0) | 73(39.3) | 1 | 1 |
| **Sedentary activity** | | | | |
| <4hour/day | 76(38.0) | 106(57.0) | 1 | 1 |
| ≥4hour/day | 124(62.0) | 80(43.0) | 2.16(1.44,3.25) | 1.75(1.10,2.79)* |
| **Stress score** | 17±4.8$^€$ | 16.6±6.0$^€$ | 1.03 (0.99,1.07) | 1.05(1.01,1.10)* |
| **BMI** | | | | |
| <25kg/m$^2$ | 77(38.5) | 92(49.5) | 1 | 1 |
| 25–29.9kg/m$^2$ | 76(38.0) | 70(37.6) | 1.30 (0.83,2.02) | 1.32(0.79,2.19) |
| ≥30kg/m$^2$ | 47(23.5) | 24(12.9) | 2.34 (1.31,4.17) | 2.82 (1.43,5.57)* |
| **Retinopathy** | | | | |
| Yes | 22(11.0) | 8(4.3) | 2.75(1.20,6.34) | 1.83(0.71,4.68) |
| No | 178(89.0) | 178(95.7) | 1 | 1 |
| **Nephropathy** | | | | |
| Yes | 22(11.0) | 14(7.5) | 1.52 (0.75,3.06) | 0.79(0.34,1.82) |
| No | 178(89.0) | 172(92.5) | 1 | 1 |
| **Serum Creatinine** | | | | |
| ≤1.1 mg/dl | 97(48.5) | 103(55.4) | 1 | 1 |
| >1.1 mg/dl | 39(19.5) | 18(9.7) | 2.30(1.23,4.29) | 2.35(1.13,4.97)* |
| not determined | 64 (32.0) | 65(34.9) | 1.05 (0.67,1.62) | 0.92(0.56,1.52) |
| **Cigarette smoking** | | | | |
| Yes | 17(8.5) | 16(8.6) | 0.98(0.48,2.01) | 1.04(0.44,2.45) |
| No | 183(91.5) | 170(91.4) | 1 | 1 |

* Statistically significant at p value ≤ 0.05

# Interaction term

¥ Median with (Q1, Q3)

€ Mean ± SD

the extracellular fluid volume initially mediates the rise in BP, despite the reduction in total peripheral resistance, leading predominantly to systolic hypertension [52]. Conversely, the association also might be due to the bidirectional nature of the relationship between creatinine and hypertension [53].

Finally, a family history of hypertension was also independently associated with hypertension among people with T2DM. The odds of cases having a family history of hypertension was about 2 times higher than their counterparts. The finding is concordant with the study conducted in Hossana NEMM hospital [26]. This is supported by the contribution of the genetic factors in the development of hypertension [54].

The use of a relatively large sample size and exhaustive inclusion of potential confounders in the model are among the strengths of this study. Further, to the best of our knowledge, this is the first study in Ethiopia to examine the association of stress and hypertension among people with T2DM. In contrast, the major limitations of this study were recall biases and chicken egg dilemma though considerable efforts were made to limit them. There might be bias towards the null and the odds ratio for the identified determinants might be underestimated. For example, patients with hypertension might have started to exercise more than those with no hypertension, which can underestimate the existing association between exercise and hypertension. The study might have also lacked power to detect an association between hypertension and rare exposures like a history of smoking.

## Conclusion

In this study, obesity, sedentary activity, stress score, the interaction of diabetes duration with insulin use, serum creatinine level, age, occupation, and family history of hypertension were identified as independent determinants of hypertension among people with T2DM. On the other hand, adherence to physical exercise, glycemic control, smoking, alcoholic drink consumption, dietary habits, self-monitoring of blood glucose, adherence with diabetes medication, diabetic education, and marital status were not associated. These findings call for the interventional strategies targeting the aforementioned determinants and suggest the clinicians be curious while deciding diabetes medications for their patients who are on follow-up. It suggests that patients with a longer duration of diabetes need more frequent and focused follow-up to prevent delay in diagnosis of a treatment failure. It also connotes further researches to examine and explain the interaction of diabetes duration with insulin use. Moreover, higher officials of the government are recommended to take interventions at governmental institutions to increase the physical activity level of the employees, and the respective patients are also recommended to compensate for reclining in sitting due to the nature of most of the governmental occupations.

## Supporting information

**S1 Dataset. "Dataset of the study".**
(XLS)

**S1 Questionnaire.**
(DOCX)

## Acknowledgments

We are grateful to Mrs. Tsion Shibiru for her valued contribution through managing the data of this study. We extend our thanks to the study participants for consenting and participating in this study.

## Author Contributions

**Conceptualization:** Kehabtimer Shiferaw Kotiso, Samson Gebremedhin, Melaku Taye, Adane Petros, Dejene Hailu.

**Data curation:** Kehabtimer Shiferaw Kotiso, Nabiha Degemu, Melaku Taye, Adane Petros, Deneke Wolde, Dejene Hailu.

**Formal analysis:** Kehabtimer Shiferaw Kotiso, Samson Gebremedhin, Fanuel Belayneh, Deneke Wolde, Dejene Hailu.

**Funding acquisition:** Kehabtimer Shiferaw Kotiso.

**Methodology:** Kehabtimer Shiferaw Kotiso, Nabiha Degemu, Samson Gebremedhin, Melaku Taye, Adane Petros, Fanuel Belayneh, Deneke Wolde, Dejene Hailu.

**Project administration:** Kehabtimer Shiferaw Kotiso, Melaku Taye.

**Software:** Kehabtimer Shiferaw Kotiso, Dejene Hailu.

**Supervision:** Kehabtimer Shiferaw Kotiso, Nabiha Degemu, Dejene Hailu.

**Validation:** Nabiha Degemu, Samson Gebremedhin, Adane Petros, Dejene Hailu.

**Writing – original draft:** Kehabtimer Shiferaw Kotiso.

**Writing – review & editing:** Kehabtimer Shiferaw Kotiso, Nabiha Degemu, Samson Gebremedhin, Melaku Taye, Adane Petros, Fanuel Belayneh, Deneke Wolde, Dejene Hailu.

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
