## [Decision Letter · Decision Letter 0]

12 May 2021

PONE-D-21-08688

Determinants of hypertension among patients with type 2 diabetes on follow-up at Tikur Anbessa Specialized Hospital, Addis Ababa: a case control study

PLOS ONE

Dear Dr. Kotiso,

Thank you for submitting your manuscript to PLOS ONE. After careful consideration, we feel that it has merit but does not fully meet PLOS ONE’s publication criteria as it currently stands. Therefore, we invite you to submit a revised version of the manuscript that addresses the points raised during the review process.

We look forward to receiving your revised manuscript.

Kind regards,

Yoshihiro Fukumoto

Academic Editor

PLOS ONE

Journal Requirements:

2. Please provide additional details regarding participant consent. In the ethics statement in the Methods and online submission information, please ensure that you have specified (1) whether consent was suitably informed and (2) what type you obtained (for instance, written or verbal). If your study included minors under age 18, state whether you obtained consent from parents or guardians. If the need for consent was waived by the ethics committee, please include this information.

4. Please include captions for ALL your Supporting Information files at the end of your manuscript, and update any in-text citations to match accordingly. Please see our Supporting Information guidelines for more information: http://journals.plos.org/plosone/s/supporting-information.

Reviewers' comments:

Reviewer's Responses to Questions

**Comments to the Author**

1. Is the manuscript technically sound, and do the data support the conclusions?

Reviewer #1: Yes

Reviewer #2: Yes

2. Has the statistical analysis been performed appropriately and rigorously? 

Reviewer #1: No

Reviewer #2: Yes

3. Have the authors made all data underlying the findings in their manuscript fully available?

Reviewer #1: Yes

Reviewer #2: Yes

4. Is the manuscript presented in an intelligible fashion and written in standard English?

Reviewer #1: Yes

Reviewer #2: Yes

5. Review Comments to the Author

Reviewer #1: This paper was a case-control study in diabetes patients in Ethiopia. The determinants of hypertension with T2DM were not physical exercise, glycemic control, smoking, alcohol intake, and education but obesity, activity, stress, duration of insulin, age, occupation, and family history of hypertension. The sample size is not so large and there was a selection bias between cases and controls. This manuscript is potentially interesting. However, there are some concerns in it.

Major comments

1) The most serious concern is a selection bias in two groups between cases and controls, I suspected that. Authors need to analysis t-test and chi-square test, and address p value in tables.

2) The second concern is not a large sample size. Authors need to mention in the section of limitation.

3) This study insisted many variables are determinants of hypertension among diabetes. What variable is the strongest or most independent factor for hypertension with diabetes? If authors insist that there was an independent association, they would need to use multiple stepwise regression analyses.

Minor comments

1) Table 1, 2, 3, and 4 were showed characteristics of patients in case and control. This manuscript have to put these tables in one.

2) There was no units for variables and no abbreviations lists in footnotes in Table 5.

3) LDL or HDL were LDL-cholesterole or HDL-cholesterol.

4) Tables were showed controls in left and cases in right. Table 6 was showed cases in left and controls in right. Tables were unified the writing.

Reviewer #2: Thank you for the opportunity to review this manuscript. The authors provided an original article assessing the determinants of hypertension in patients with diabetes mellitus at TASH. They presented the article in a meaningful way, however, the following issues need to be fixed/ clarified to make it easier for the reader to follow, and make it methodologically sound.

1) Title: please specify the type of diabetes mellitus considered in the cohort (type 2 DM as indicated in the subsequent sections), including an addition of ‘mellitus’ in the phrase. Also, use a hyphen in ‘case control’.

2) Abstract: the phrase ‘type 2 diabetic patients’ needs revision, such as ‘patients with T2DM’.

3) Introduction: line 34: check, and revise references 23 and 24.

Line 36: write the full form of WHO

4) Methods and materials: there is an inconsistency in the use of the terms; T2DM and/or type 2 DM. Please revise and apply one uniformly across the manuscript.

The specific scheduled date of study period needs to be defined.

The sample size estimation is barely defined which requires major revision. It is difficult to follow authors’ assumption (reference) of exposure variables among the cases and controls considered in the computation. As both groups emerge from a DM cohort, and that DM cannot, independently, predict hypertension among the cases only, the figure (OR=2.28) in line70, and how they came up with the stated sample size is still unclear.

Sampling method of this study is not clear. For example, in lines 45-47, of the introduction summary, the authors indicated that participant selection flaws were a problem in earlier studies. As cases and controls are not apparent to researchers from the outset, using simple random sampling simply cannot be possible. It is imperative that authors clearly indicate the procedure showing how cases and controls were recruited.

5) Results: in line 155; ‘two-thirds’ for 62.4% appears unsound, and needs revision.

Please check and revise table 5.

Lines 203-204: the term ‘diabetic medications’, also requires to be rephrased.

6) Discussion: well discussed, however, the use of abbreviations for the first time should be avoided.

Generally, the authors used some abbreviations throughout the paper without first presenting the full form. Authors are advised to have a look at on this terms, and fix for clarity.

6. PLOS authors have the option to publish the peer review history of their article (what does this mean?). If published, this will include your full peer review and any attached files.

Reviewer #1: No

Reviewer #2: No

---

## [Author Response · Author response to Decision Letter 0]

14 Jul 2021

Dear reviewers and editor, 

Thank you for your review and important issues you raised in the paper. 

We have read the reviewer's comments carefully and hope that the revised version now submitted will be regarded as having enhanced the previous version. The authors very welcome the reviewer's comments and suggestions. These contributions have appreciably improved the final paper quality. Hereunder, kindly get the responses to the points raised under each point.

Response to editor

We've checked your submission and before we can proceed, we need you to address the following issues:

- Please include a legend for figure 1.

BMI in the figure 1. was written in full in the revised figure, and the title of the figure was also included in the manuscript to locate the position of the figure.

Response to reviewers

Reviewers' comments:

Reviewer's Responses to Questions

Comments to the Author

1. Is the manuscript technically sound, and do the data support the conclusions?

Reviewer #1: Yes

Reviewer #2: Yes

2. Has the statistical analysis been performed appropriately and rigorously?

Reviewer #1: No

Reviewer #2: Yes

3. Have the authors made all data underlying the findings in their manuscript fully available?

Reviewer #1: Yes

Reviewer #2: Yes

4. Is the manuscript presented in an intelligible fashion and written in standard English?

Reviewer #1: Yes

Reviewer #2: Yes

5. Review Comments to the Author

Reviewer #1: This paper was a case-control study in diabetes patients in Ethiopia. The determinants of hypertension with T2DM were not physical exercise, glycemic control, smoking, alcohol intake, and education but obesity, activity, stress, duration of insulin, age, occupation, and family history of hypertension. The sample size is not so large and there was a selection bias between cases and controls. This manuscript is potentially interesting. However, there are some concerns in it.

Major comments

1) The most serious concern is a selection bias in two groups between cases and controls, I suspected that. Authors need to analysis t-test and chi-square test, and address p value in tables.

Comment: The detail procedures of selecting the participants were added in the revised manuscript based on the reviewer’s comment.

To eliminate selection bias, we randomly selected both the cases and controls using the digital record of the hospital, where the status and care of DM patients are regularly recorded. Because of the availability of digital record in the hospital, where the status and care of DM patients are regularly recorded per each visit to the diabetic clinic, we were able to generate sample frame from it for both cases and controls. For e.g. whether the patient has comorbidity including hypertension or not is recorded in the digital record, and all DM patients be screened for hypertension per each visit. As a result, those T2DM patients already recorded as hypertensive in the record were considered as list of cases, and the non-hypertensives as controls. In the process, the newly diagnosed cases were also managed accordingly.

Comment: Bivariable binary logistic regression analysis was conducted instead of t-test and chi-square test for each variable (since the model can handle both continuous and categorical variables), and variables with a p-value of <0.25 were presented in table.

2) The second concern is not a large sample size. Authors need to mention in the section of limitation.

Comment: Thank you for the issue you raised associated with the sample size. We stated under limitation that “The study might have also lacked power to detect an association between hypertension and rare exposures like a history of smoking.” to address the issue raised by the reviewer.

3) This study insisted many variables are determinants of hypertension among diabetes. What variable is the strongest or most independent factor for hypertension with diabetes? If authors insist that there was an independent association, they would need to use multiple stepwise regression analyses.

Comment: In terms of measures of association (Odds ratio), obesity was the strongest independent factor of hypertension. However, it’s not to mean that variables (particularly the countinous one, such as age and stress score) with the least odds ratio are the weakest independent factor, because the odds ratio for these variables is difficult to be compared with the categorical variables for they have different interpretation. On another note, we have used enter method rather than the stepwise method for the analysis both because of the limitations associated with stepwise (as supposed by different statisticians) and the AIC signified the enter method as a best fitting model for our data as stated under “Data processing and analysis”

Minor comments

1) Table 1, 2, 3, and 4 were showed characteristics of patients in case and control. This manuscript have to put these tables in one.

Comment: Amendments were made based on the reviewer’s comment. 

2) There was no units for variables and no abbreviations lists in footnotes in Table 5.

Comment: Units were added, and abbreviation lists were put in footnotes in the revised manuscript.

3) LDL or HDL were LDL-cholesterole or HDL-cholesterol.

Comment: Corrected based on the comment.

4) Tables were showed controls in left and cases in right. Table 6 was showed cases in left and controls in right. Tables were unified the writing.

Comment: Corrected according to the comment.

Reviewer #2: Thank you for the opportunity to review this manuscript. The authors provided an original article assessing the determinants of hypertension in patients with diabetes mellitus at TASH. They presented the article in a meaningful way, however, the following issues need to be fixed/ clarified to make it easier for the reader to follow, and make it methodologically sound.

1) Title: please specify the type of diabetes mellitus considered in the cohort (type 2 DM as indicated in the subsequent sections), including an addition of ‘mellitus’ in the phrase. Also, use a hyphen in ‘case control’.

Comment: Corrected based on the comment.

2) Abstract: the phrase ‘type 2 diabetic patients’ needs revision, such as ‘patients with T2DM’.

Comment: Revised based on the comment.

3) Introduction: line 34: check, and revise references 23 and 24.

Comment: Checked and revised based on the reviewer’s comment.

Line 36: write the full form of WHO

Comment: WHO was written in full in the revised manuscript based on the reviewer’s comment.

4) Methods and materials: there is an inconsistency in the use of the terms; T2DM and/or type 2 DM. Please revise and apply one uniformly across the manuscript.

Comment: Revised based on the comment

The specific scheduled date of study period needs to be defined.

Comment: The specific scheduled date was stated in the revised manuscript.

The sample size estimation is barely defined which requires major revision. It is difficult to follow authors’ assumption (reference) of exposure variables among the cases and controls considered in the computation. As both groups emerge from a DM cohort, and that DM cannot, independently, predict hypertension among the cases only, the figure (OR=2.28) in line70, and how they came up with the stated sample size is still unclear.

Comment: Sorry for the mistake made while editing for language. It was mistakenly written as “Proportion of controls” but now corrected as “proportion of controls with exposure.” The command that we used in stata to compute our sample size was “power twoproportions (0.4 (0.025)0.7), test(lrchi2) oratio(2.28) power(0.9) table.” To obtain larger sample size, we used a range for the proportion of controls with exposure ranging from 0.4 to 0.7, based on the pre-study estimate of 0.56. Finally, the proportion we used to estimate the sample size was 0.7 because of yielding the largest sample size.

Sampling method of this study is not clear. For example, in lines 45-47, of the introduction summary, the authors indicated that participant selection flaws were a problem in earlier studies. As cases and controls are not apparent to researchers from the outset, using simple random sampling simply cannot be possible. It is imperative that authors clearly indicate the procedure showing how cases and controls were recruited.

Comment: Flawed participant population selection was stated in the introduction summary to explain limitations of the previous studies due to the type of population participated in the studies which might bias and or confound the results (population selection rather than the sampling technique). To mention some of the major limitations, participants in whom HTN diagnosis precedes DM diagnosis were not excluded in the previous studies (Tadesse, Amare et al. 2018, Kotiso, Mekebo et al. 2020) regardless of the fact that this might contaminate the results by mixing-up factors determining for the development of DM in HTN patients (in whom the risk factors for HTN might be similar with that of the general population) rather than HTN in DM. Besides, the other study (Mariye, Girmay et al. 2019) included all DM patients which might also bias and confound the results unless properly controlled because of similarity in the pathophysiology of T2DM with that of HTN, and T2DM accounting the vast majority of DM population. Thus, this study tried to address the aforementioned issues to provide more meaningful and valid information. 

Because of the availability of digital record in the hospital, where the status and care of DM patients are regularly recorded per each visit to the diabetic clinic, we were able to generate sample frame from it for both cases and controls. For e.g. whether the patient has comorbidity including hypertension or not is recorded in the digital record, and all DM patients be screened for hypertension per each visit. As a result, those T2DM patients already recorded as hypertensive in the record were considered as list of cases, and the non-hypertensives as controls. In the process, the newly diagnosed cases were also managed accordingly.

The detail procedures of sampling were added in the revised manuscript based on the reviewer’s comment.

5) Results: in line 155; ‘two-thirds’ for 62.4% appears unsound, and needs revision.

Comment: It was addressed in the revised manuscript.

Please check and revise table 5.

Comment: table 5 was revised.

Lines 203-204: the term ‘diabetic medications’, also requires to be rephrased.

Comment: The phrase diabetic medication was rephrased as diabetes medication.

6) Discussion: well discussed, however, the use of abbreviations for the first time should be avoided.

Generally, the authors used some abbreviations throughout the paper without first presenting the full form. Authors are advised to have a look at on this terms, and fix for clarity.

Comment: All the abbreviation issues are fixed in the revised manuscript.

Thank you for your consideration. We look forward to hearing from you.

Sincerely,

The authors

---

## [Decision Letter · Decision Letter 1]

31 Jul 2021

PONE-D-21-08688R1

Determinants of hypertension among patients with type 2 diabetes mellitus on follow-up at Tikur Anbessa Specialized Hospital, Addis Ababa: a case-control study

PLOS ONE

Dear Dr. Kotiso,

Thank you for submitting your manuscript to PLOS ONE. After careful consideration, we feel that it has merit but does not fully meet PLOS ONE’s publication criteria as it currently stands. Therefore, we invite you to submit a revised version of the manuscript that addresses the points raised during the review process.

We look forward to receiving your revised manuscript.

Kind regards,

Yoshihiro Fukumoto

Academic Editor

PLOS ONE

Journal Requirements:

Reviewers' comments:

Reviewer's Responses to Questions

**Comments to the Author**

1. If the authors have adequately addressed your comments raised in a previous round of review and you feel that this manuscript is now acceptable for publication, you may indicate that here to bypass the “Comments to the Author” section, enter your conflict of interest statement in the “Confidential to Editor” section, and submit your "Accept" recommendation.

Reviewer #1: All comments have been addressed

Reviewer #2: All comments have been addressed

2. Is the manuscript technically sound, and do the data support the conclusions?

Reviewer #1: (No Response)

Reviewer #2: Yes

3. Has the statistical analysis been performed appropriately and rigorously? 

Reviewer #1: (No Response)

Reviewer #2: Yes

4. Have the authors made all data underlying the findings in their manuscript fully available?

Reviewer #1: (No Response)

Reviewer #2: Yes

5. Is the manuscript presented in an intelligible fashion and written in standard English?

Reviewer #1: (No Response)

Reviewer #2: Yes

6. Review Comments to the Author

Reviewer #1: In table 2, units of mean SBP and DBP were mm Hg. This units need no space between mm and Hg. 'mmHg'

In table 3, AOR in DM duration was 1.035(1.01, 1.07). AOR was needed to unified two alignment.

Reviewer #2: Thank you again for presenting the revised form of your work. The authors have revised the manuscript incorporating comments suggested. I advise the inclusion of 'materials' in the methods sub-section of the abstract. Also please check the acknowledgement section for patients, while offering their full consent and participation, should have been acknowledged, or if that has been addressed elsewhere.

7. PLOS authors have the option to publish the peer review history of their article (what does this mean?). If published, this will include your full peer review and any attached files.

Reviewer #1: No

Reviewer #2: No

---

## [Author Response · Author response to Decision Letter 1]

31 Jul 2021

Dear reviewers, 

Thank you for your review and important issues you raised in the paper. 

We have read the reviewer's comments carefully and hope that the revised version now submitted will be regarded as having enhanced the previous version. The authors very welcome the reviewer's comments and suggestions. These contributions have appreciably improved the final paper quality. Hereunder, kindly get the responses to the points raised under each point.

Response to reviewer #1:

Review Comments to the Author

Reviewer #1: In table 2, units of mean SBP and DBP were mm Hg. This units need no space between mm and Hg. 'mmHg'

Comment: the spaces were removed and corrected as mmHG based on the reviewer’s comment. 

In table 3, AOR in DM duration was 1.035(1.01, 1.07). AOR was needed to unified two alignment.

Comment: the decimal place was rounded to two places based on the reviewer’s comment.

Response to Reviewer #2:

Reviewer #2: Thank you again for presenting the revised form of your work. The authors have revised the manuscript incorporating comments suggested. I advise the inclusion of 'materials' in the methods sub-section of the abstract. Also please check the acknowledgement section for patients, while offering their full consent and participation, should have been acknowledged, or if that has been addressed elsewhere.

Comment: materials was now included in the abstract subsection methods, and the study participants were also acknowledged in the revised version based on the reviewer’s comment.

Thank you for your consideration. We look forward to hearing from you.

Sincerely,

The corresponding author

---

## [Decision Letter · Decision Letter 2]

6 Aug 2021

Determinants of hypertension among patients with type 2 diabetes mellitus on follow-up at Tikur Anbessa Specialized Hospital, Addis Ababa: a case-control study

PONE-D-21-08688R2

Dear Dr. Kotiso,

We’re pleased to inform you that your manuscript has been judged scientifically suitable for publication and will be formally accepted for publication once it meets all outstanding technical requirements.

Kind regards,

Yoshihiro Fukumoto

Academic Editor

PLOS ONE

Additional Editor Comments (optional):

Reviewers' comments:

Reviewer's Responses to Questions

**Comments to the Author**

1. If the authors have adequately addressed your comments raised in a previous round of review and you feel that this manuscript is now acceptable for publication, you may indicate that here to bypass the “Comments to the Author” section, enter your conflict of interest statement in the “Confidential to Editor” section, and submit your "Accept" recommendation.

Reviewer #1: All comments have been addressed

Reviewer #2: All comments have been addressed

2. Is the manuscript technically sound, and do the data support the conclusions?

Reviewer #1: Yes

Reviewer #2: Yes

3. Has the statistical analysis been performed appropriately and rigorously? 

Reviewer #1: Yes

Reviewer #2: Yes

4. Have the authors made all data underlying the findings in their manuscript fully available?

Reviewer #1: Yes

Reviewer #2: Yes

5. Is the manuscript presented in an intelligible fashion and written in standard English?

Reviewer #1: Yes

Reviewer #2: Yes

6. Review Comments to the Author

Reviewer #1: (No Response)

Reviewer #2: I have found the paper meaningful and with potential contribution to practice. I have no objection to its present form thus far.

7. PLOS authors have the option to publish the peer review history of their article (what does this mean?). If published, this will include your full peer review and any attached files.

Reviewer #1: No

Reviewer #2: No

---

## [Editor Report · Acceptance letter]

13 Aug 2021

PONE-D-21-08688R2 

Determinants of hypertension among patients with type 2 diabetes mellitus on follow-up at Tikur Anbessa Specialized Hospital, Addis Ababa: a case-control study 

Dear Dr. Kotiso:

I'm pleased to inform you that your manuscript has been deemed suitable for publication in PLOS ONE. Congratulations! Your manuscript is now with our production department. 

Kind regards, 

on behalf of

Dr. Yoshihiro Fukumoto 

Academic Editor

PLOS ONE